# Emerging Therapeutic Strategies to Overcome Drug Resistance in Multiple Myeloma

**DOI:** 10.3390/cancers13071686

**Published:** 2021-04-02

**Authors:** Lorraine N. Davis, Daniel W. Sherbenou

**Affiliations:** 1Division of Hematology, Department of Medicine, University of Colorado Anschutz Medical Campus, Aurora, CO 80045, USA; lorraine.davis@cuanschutz.edu; 2Department of Blood Disorders and Cell Therapies, University of Colorado Comprehensive Cancer Center, Aurora, CO 80045, USA

**Keywords:** multiple myeloma, drug resistance, resistance mechanisms, immunomodulatory drugs, proteasome inhibitors, monoclonal antibodies, immunotherapy, treatment, clinical trials

## Abstract

**Simple Summary:**

Multiple myeloma is a deadly blood cancer, but fortunately drug development has substantially prolonged the lifespan of patients to average more than a decade after diagnosis with optimal therapy. As a result, the population of patients living with multiple myeloma has grown considerably. Through its course, patients suffer repeated relapses for which they require new lines of treatment. Currently, the key drug classes for treatment are immunomodulatory drugs, proteasome inhibitors, and monoclonal antibodies. The goal of this review is to summarize the understanding of the problem of resistance to these drugs, which is ultimately responsible for patient fatality. In addition, we will focus on how new agents that are promising in clinical trials overcome resistance.

**Abstract:**

Multiple myeloma is a malignant plasma cell neoplasm that remains incurable and is ultimately fatal when patients acquire multi-drug resistance. Thus, advancing our understanding of the mechanisms behind drug resistance in multi-relapsed patients is critical for developing better strategies to extend their lifespan. Here, we review the understanding of resistance to the three key drug classes approved for multiple myeloma treatment: immunomodulatory drugs, proteasome inhibitors, and monoclonal antibodies. We consider how the complex, heterogenous biology of multiple myeloma may influence the acquisition of drug resistance and reflect on the gaps in knowledge where additional research is needed to improve our treatment approaches. Fortunately, many agents are currently being evaluated preclinically and in clinical trials that have the potential to overcome or delay drug resistance, including next-generation immunomodulatory drugs and proteasome inhibitors, novel small molecule drugs, chimeric antigen receptor T cells, antibody-drug conjugates, and bispecific antibodies. For each class, we discuss the potential of these strategies to overcome resistance through modifying agents within each class or new classes without cross-resistance to currently available drugs.

## 1. Introduction

Multiple myeloma (MM) is a malignant plasma cell (PC) neoplasm which is controllable initially with treatment, but ultimately the development of drug resistant disease leads to patient mortality. MM afflicts more than 30,000 Americans each year and is the second most prevalent hematologic malignancy [1]. Over the last decade the incidence of MM has continued to increase, with an estimated 12,830 deaths predicted in 2020 [1]. MM is characterized by the clonal proliferation of plasma cells, predominantly in the bone marrow, associated with overproduction of monoclonal immunoglobulin (Ig) known as M-protein. Complications of MM include bone lesions, hypercalcemia, renal failure, cytopenias, and immune suppression [2]. Treatment options for MM have significantly improved over the past two decades with the emergence of many new anti-myeloma drugs, increasing the average life expectancy from three-to-four years to seven-to-eight years for standard-risk disease [2]. However, patients still suffer repeated relapses and inevitably develop multi-drug resistance.

Multiple myeloma is a complex and heterogeneous disease. The underlying genetic abnormalities that initiate and promote MM are extraordinarily diverse. Cytogenetic abnormalities in multiple myeloma that confer poorer outcomes include translocations t (4;14), t (14;16), and t (14;20), deletion of chromosome 17p and gain of chromosome 1q. In addition, accumulation of genetic mutations increases over the disease course and occurs more rapidly in cytogenetically high-risk patients [3]. However, although these genetic abnormalities have prognostic value, they do not correlate with initial disease response. Recently, subclonal heterogeneity has become appreciated to occur in MM and tends to evolve with disease progression through selection of more drug-resistant or aggressive subclones [4,5]. Spatial heterogeneity adds further complexity, as subclones can vary in distribution across simultaneous biopsy sites [6]. Although clonal evolution parallels drug resistance development, how these are mechanistically linked remains unclear. Irrespective of genetic risk factors, multi-drug resistance development is a critical issue, as patients who are triple-class refractory to proteasome inhibitors (PIs), immunomodulatory drugs (IMiDs), and anti-CD38 monoclonal antibodies have only 5.6 months median overall survival [7]. Consequently, triple-refractory patients are in dire need of new therapies which can circumvent triple-class resistance.

Acquired drug resistance, leading to triple-class refractory MM, is the root cause of relapsed disease and remains the greatest obstacle to prolonging patients’ lives. With many drugs available, patients are cycled through combinations of two to four agents at a time. These regimens include at least sixteen individual agents in seven major drug classes. This diverse toolbox has created a complex situation where patients develop drug resistance to the available agents in a gradual manner that varies considerably from patient-to-patient. Advancing our understanding of multi-drug resistance is critical for the development of more effective strategies for the treatment of relapsed/refractory MM (RRMM). This starts with clear characterization of the underlying mechanisms of resistance to each drug class. In this review, we focus on the development of drug resistance to the three key therapeutic classes in MM treatment: Immunomodulatory Drugs (IMiDs), Proteasome Inhibitors (PIs), and immunotherapies (Table 1). We will also touch on the resistance to steroids and chemotherapies. We summarize the current knowledge of the genetic, molecular, and cellular mechanisms of resistance. To develop a complete picture of the current state of MM drug resistance, we will also consider lessons from ongoing clinical trials and examine how current drug development for MM may overcome drug resistance and improve the bleak outcome of triple-class refractory MM.

## 2. Immunomodulatory Drugs

IMiDs have revolutionized MM treatment in the past 20 years. The first IMiD, thalidomide, was initially used as an antiemetic for pregnant women, but unfortunately had teratogenic effects. Serendipitously, it was later discovered to be cytotoxic to MM cells and was repurposed for MM treatment [29]. Due to some undesirable side effects, thalidomide has largely been superseded by more efficacious and tolerable derivatives. Chemical modification led to second generation lenalidomide, and structural elements of both thalidomide and lenalidomide were combined to form pomalidomide, the most potent of the IMiDs to be FDA-approved [30]. Lenalidomide is a key part of standard of care drug combinations in the diagnosis setting and as maintenance [31]. In RRMM, multiple lenalidomide and pomalidomide-based regimens are FDA-approved.

### 2.1. IMiD Mechanism of Action

Only relatively recently did the unique anti-myeloma mechanism of the IMiDs come to light. IMiDs bind to Cereblon (CRBN), a protein which dictates the substrate specificity of the CRL4^CRBN^ E3 ubiquitin ligase [32,33]. IMiD binding to CRBN alters its substrate affinity and leads CRL4^CRBN^ to target a new set of substrates for proteasomal degradation [34]. The key neosubstrates in MM cells are Ikaros (IKZF1) and Aiolos (IKZF3) [35,36]. These transcription factors (TFs) regulate cell fate decisions in normal lymphopoiesis and PC development [37]. IKZF1 and IKZF3 upregulate the TFs interferon regulatory factor 4 (IRF4) and c-MYC (MYC), which form a positive autoregulatory loop necessary for MM survival and proliferation [38]. These four TFs can be collectively referred to as the Ikaros axis, and knockdown experiments support that IRF4 and MYC dependence is the means by which IMiDs ultimately exert their activity [38,39]. In summary, IMiDs act through promoting the rapid degradation of IKZF1 and IKZF3 via CRBN-dependent ubiquitination, leading to downregulation of IRF4 and MYC (Figure 1).

### 2.2. IMiD Resistance

Most patients are initially sensitive to IMiDs. First tested in the RRMM setting, single-agent lenalidomide showed an overall response rate (ORR; ≥50% decrease in disease burden) of 26% [40]. Based on the IMiD mechanism of action, perturbations in the CRL4^CRBN^ complex and Ikaros axis are logical candidate drivers of resistance (Figure 1A). This hypothesis was initially supported with MM cell lines, as targeted disruption of CRBN and IKZF1/3 binding yielded IMiD resistance [35,36,41]. Later, a study of primary samples showed CRBN mutations in 12% (6/50) of RRMM patients, 88% of which were clinically IMiD-refractory. These mutations were not present in matched pre-treatment samples, suggesting these were acquired in response to treatment [11]. Similar studies have also showed acquired CRBN mutations in RRMM patients [10,42]. With lower frequencies, mutations in IKZF1, IRF4, and the CUL4B component of CRL4^CRBN^ have also been reported [10,11]. Although CRBN mutations are relative rare, separate studies have found CRBN downregulation after lenalidomide treatment in 89% (8/9) of patients at the mRNA level and 71% (5/7) of patients at the protein level [12,41]. Similarly, high CRBN mRNA expression has been associated with IMiD responsiveness [43,44,45,46], and IMiD-resistant MM cell lines reflect this at the protein level [8,9,32]. CRBN protein levels are affected by the SCF^Fbxo7^ E3 ubiquitin ligase, which is regulated by the CSN signalosome [15]. Studies have also looked at IKZF1/3, but conflicting results have emerged as to whether their expression correlates with IMiD response [45,47,48]. Collectively, mutations in the CRBN-Ikaros axis exist in IMiD resistant patients, but the patients tested thus far suggest that CRBN downregulation may a more commonly acquired event. 

Additional modulators of CRBN or IKZF1/3 activity may also contribute to resistance. This can occur if IKZF1/3 degradation is prevented through interference of binding with CRBN (Figure 1B). Zhou and colleagues found that RUNX1/3 compete with CRBN for binding and prevent IKZF1/3 degradation [14]. In MM cell lines and patient samples (*n* = 5), inhibiting RUNX1/3 reversed lenalidomide resistance. CRBN also has several substrates that if upregulated may outcompete IKZF1/3 for CRBN binding in the presence of IMiDs [8]. Interestingly, Eichner et al. found that CRBN also functions as a chaperone for MCT1 and CD147, and IMiD binding leads to mislocalization of those proteins [16]. This study also found that lenalidomide-resistant cell lines maintained CD147 and MCT1 levels in response to IMiDs. In aggregate, these studies clearly show the complexity of CRBN biology, but their IMiD resistance mechanisms have only been characterized in cell lines to this point and still need to be investigated in patient samples to fully assess clinical relevance. While IKZF1/3 and CRBN have been studied more extensively, investigating the downstream effects on IRF4 and MYC will also be important in determining the functional consequence of these mechanisms in clinical IMiD resistance.

Other IMiD resistance mechanisms could bypass CBRN and the Ikaros axis to promote MM cell survival (Figure 1C). Although low CRBN expression tends to correlate with IMiD resistance, some patients with similarly high CRBN levels have differential drug sensitivity and many IMiD-resistant MM cases do not show abnormalities in CRBN, IKZF1 or IKZF3 [11,44]. The activation of other signaling pathways may compensate, potentially through upregulating IRF4, MYC, or other unrelated pro-survival factors. Consistent with this, Zhu and colleagues found that lenalidomide-adapted MM cell lines lacking CRBN abnormalities showed impaired IRF4 downregulation and upregulation of IL-6/STAT3 signaling [9]. In patients, high IL-6 expression was associated with shorter responses [9]. Downstream, IL-6/STAT3 signaling may maintain IRF4 and/or MYC upregulation and induce pro-survival MAPK and PI3K pathway activation [9,13,17]. Based on these findings, investigating IL-6 antagonists and STAT3 inhibitors in the context of IMiD-refractory patients is warranted.

In sum, the evidence thus far supports the notion that IMiD resistance occurs mostly by mutation or gene expression changes in the CRBN-Ikaros axis or by compensatory growth pathways. Sequencing studies have found low mutation rates in the CRBN-Ikaros axis in the RRMM setting, whereas clonal evolution with mutations in other proliferative pathways have been comparatively common [4,10,11]. Currently, it remains unclear from to what degree RRMM patients’ disease remains Ikaros dependent [49]. The clinical trials discussed next have shown that IKZF1/3 remain valid targets in at least some IMiD-exposed patients.

### 2.3. Overcoming IMiD Resistance

Next-generation CRBN E3 Ligase Modulators (CELMoDs) show higher potency and may be able to overcome IMiD resistance in Ikaros axis-dependent MM. The CELMoDs iberdomide, avadomide, and CC-92480 are currently in clinical development for RRMM (Table 2). Iberdomide and avadomide have 20-fold higher binding affinity for CRBN compared to lenalidomide and increased potency of IKZF1/3 degradation [29,38]. CELMoDs overcome IMiD resistance to some degree in MM cell lines, likely through accelerating the degradation of IKZF1/3 [8,50,51]. CELMoDs are currently being evaluated as for treatment of RRMM (Table 2). In a Phase 1b/2a dose escalation trial of IMiD exposed patients, the ORR for iberdomide with dexamethasone was 31% [52]. The promising results with iberdomide show that there remains a significant portion of patients whose IMiD resistance can be overcome with more potent IKZF1/3 degradation. To this point, avadomide has only been reported in two patients with RRMM thus far, neither of which responded [53]. Clinical results are not yet available with CC-92480. In the cases where IMiD resistance results from CRL4^CRBN^ component or Ikaros mutations, these agents may be ineffective and thus studying the relationship between mutational status and clinical response will be helpful in the development of CELMoDs. The direct ex vivo measurement of IMiD response may be a valuable way to determine when CELMoDs are likely to benefit an IMiD-refractory patient [54].

On the other hand, targeting alternative pathways will likely be needed if patients’ disease become Ikaros axis independent. This could occur by MYC decoupling from IKZF1/3 downregulation by IMiDs, making other means of MYC inhibition attractive. Bromodomain extra-terminal (BET) inhibitors are one possibility, as they downregulate MYC transcription. The BET inhibitor JQ1 has shown efficacy in both IMiD sensitive and resistant MM cell lines [68]. The MYC-MAX dimerization inhibitor 10058-F4 also displayed anti-myeloma effects in MM cell lines [69]. Other MYC inhibitors have also shown promise as anti-cancer therapeutics, but these have yet to be evaluated in MM [70]. Further study of these approaches is merited in IMiD-resistant MM.

## 3. Proteasome Inhibitors

Proteasome inhibitors have been a standard-of-care in MM treatment over the past two decades. Bortezomib is a first-in class proteasome inhibitor (PI) that is used in both the newly diagnosed and RRMM setting. Newer-generation PIs carfilzomib and orally bioavailable ixazomib are FDA-approved for combination therapy in the RRMM setting.

### 3.1. PI Mechanism of Action

Approximately 90% of protein turnover occurs through the ubiquitin-proteasome system [71]. Since MM cells are highly secretory, they are exceptionally dependent on their ability to dispose of misfolded and aggregated proteins via proteasomal degradation. Consistent with this, normal PCs and malignant MM cells are heavily reliant on the unfolded protein response (UPR) to ensure proper protein folding and turnover [71]. Proteasomes are proteolytic complexes that degrade ubiquitylated proteins and are composed of a 20S core catalytic particle and 19S regulatory particle. There are three distinct catalytic sites within the 20S particle: the chymotrypsin-like site in the β5 subunit, trypsin-like site in the β2 subunit, and caspase-like site in the β1 subunit [72]. PIs are short peptide-based structures that bind to the catalytic sites. Bortezomib and ixazomib bind reversibly to the chymotrypsin-like site. In contrast, carfilzomib binds irreversibly and with much higher affinity to the β5 active site and also binds with lower affinity to the trypsin-like and caspase-like active sites [72,73]. Proteasome inhibition results in an excessive accumulation of proteins within MM cells, resulting in prolonged, irresolvable ER stress, and apoptosis (Figure 2) [74,75].

### 3.2. PI Resistance

The proteasome, as the direct drug class target, is a well-studied candidate in acquired PI resistance (Figure 2A). Based on the binding site shared between bortezomib and carfilzomib, PSMB5 mutations in the β5 subunit were predicted be a hotspot for PI resistance, and bortezomib-adapted MM cell lines have shown mutations there [76,77]. In MM patient specimens, proteasome mutations have been extensively characterized by Barrio and colleagues. Mutations in PSMB5 occurred at a frequency of 1.1% (4/355) in relapsed patients, compared to 0.08% (1/1, 241) in newly diagnosed patients [19]. Among those characterized, bortezomib resistance was observed and carfilzomib resistance was a variable. Mutations outside the catalytic core also occur at low frequencies in 20S subunit genes PSMA1, PSMB8, and PSMB9, the 19S subunit gene PSMD1, and the proteasome assembly chaperone PSMG2, but their functional consequences are unknown [11,19]. While it was postulated that proteasome overexpression may mediate bortezomib resistance, a study of bortezomib-resistant primary MM found that most proteasome subunits, including PSMB5, were downregulated [18]. In addition, low 19S regulatory subunit gene expression appears to correlate with PI non-responsiveness [20,78]. Conversely, the 20S subunit PSMA3 was upregulated in bortezomib-resistant patients and conferred resistance when overexpressed in MM cell lines [79]. Thus, proteasome subunit expression changes in may contribute to PI resistance, but the frequency whereby this occurs in patients are not clear and warrant further investigation.

Resistance mechanisms independent of the proteasome itself have also been supported in MM. Since PIs rely on disrupting proteostasis for their cytotoxic effects, resistance may occur through shifting to a less secretory state that can withstand UPS disruption. X-box-protein 1 (XBP1) is critical for PC differentiation and is a master UPR regulator. XBP1 is spliced into its active form (Xbp1s) by Inositol-requiring enzyme 1α (Ire1α), which allows PCs and MM cells to maintain proteostasis [80]. Leung-Hagesteijn and colleagues showed IRE1α-Xbp1 pathway downregulation in PI-resistant patient samples [18]. In line with this, populations of bortezomib-resistant primary MM cells possessed a pre-plasmablast phenotype, and increased fractions of less differentiated cells have been observed following bortezomib-based regimens [18,81,82]. This switch to a less differentiated and secretory, state suggests that proteostasis disruption may no longer be catastrophic to resistant MM cells (Figure 2B). Another consequence of PI-induced ER stress is the overproduction of reactive oxygen species (ROS) and oxidative stress [83]. However, PI-resistant MM cells may rewire their metabolism to better handle ROS (Figure 2C) [84,85]. In addition to UPR, PCs also rely on autophagy for protein disposal [56,86]. Not surprisingly, increased autophagy has been observed in MM cell line acquired PI resistance (Figure 2D) [22,56,87,88]. Overall, it appears MM cells may evade PI-induced cytotoxicity through changes in cell state or compensatory pathways, suggesting novel therapeutic strategies that target PI-resistant cellular features or further exacerbate proteotoxic stress may be required.

A more generic pharmacokinetic mechanism of PI resistance may be increased drug efflux (Figure 2E). Drug efflux pumps, or ATP-binding cassette (ABC) transporters, shuttle drugs and their metabolites from the intracellular to the extracellular space [89]. Increased drug efflux leads to resistance by preventing drugs from interacting with their intracellular targets. Carfilzomib and bortezomib are substrates of the multidrug transporter P-glycoprotein (MDR1) [89,90]. MDR1 is upregulated in response to PI treatment and was also identified in PI-adapted MM cell lines [23,91]. MDR1 inhibition re-sensitizes MM cells to PIs in vitro, but clinically showed too many side effects and no benefit [92]. Thus, a different approach is needed. Nima-related protein kinase 2 (Nek2) controls many ABC pathways and has been implicated in PI resistance through several studies [24,91,93]. Pharmacologic inhibition of Nek2 reversed bortezomib resistance in MM xenograft models [21], suggesting this strategy may impede increased drug efflux in PI-refractory MM patients.

### 3.3. Overcoming PI Resistance

Many drugs are being investigated for MM treatment that may overcome PI resistance. Within the same drug class, the novel pan-proteasome inhibitor marizomib (NPI-0052) underwent clinical trials for RRMM. Marizomib irreversibly binds to all three proteolytic sites of the 20S proteasome (β5, β2, β1) and binds to the β5 site with two times higher potency compared to bortezomib [55,94]. Marizomib has activity in MM cells with acquired bortezomib resistance, but it has not been tested in carfilzomib resistant cells [94]. While Phase I and II clinical trials were completed for marizomib for RRMM in 2017, at this point there has been no further clinical development [55].

Several proteasome-independent strategies for inducing proteotoxic stress are being pursued for the PI refractory setting. Kinase inhibitors of IRE1α have shown cytotoxicity in MM cell lines, xenograft models and patient samples [80]. While not yet directly assessed with PI resistance, inhibiting IRE1α similarly triggers ER stress. Attacking another node of proteostasis, the E1 ubiquitin-activating enzyme inhibitor TAK-243 (MLN7243) overcame PI resistance in MM cell line models and RRMM primary samples, inducing all three ER stress pathways [95]. TAK-243 is currently in Phase I trials for myeloid malignancies and hopefully will be assessed in MM soon. Another potential target in PI-refractory disease is the Vasolin-containing protein (p97), which delivers ubiquitin-tagged proteins to the proteasome. The p97 inhibitor CB-5083 induces UPR in MM cells and demonstrated efficacy in PI-resistant patient samples and preclinical models [96,97]. Unfortunately, clinical trials for CB-5083 were terminated due to off-target effects, so this strategy may require development of less toxic inhibitors [97]. The pan-histone deacetylase (HDAC) inhibitor panobinostat is approved for RRMM treatment, and preclinical studies supported its synergy with bortezomib through inhibiting the aggresome pathway [98]. However, panobinostat is infrequently utilized in MM due to associated toxicities [99]. The new HDAC inhibitor ricolinostat is more selective and possesses less off-target effects. Ricolinostat showed activity in carfilzomib-refractory patients and is currently in a Phase I/II trial for RRMM [65]. Autophagy inhibitors are also being explored. The lysosomal inhibitor hydroxychloroquine enhanced carfilzomib effects in cell lines and is being evaluated in a Phase I trial for RRMM treatment in combination with bortezomib (Table 2) [56]. Lastly, protein translation inhibition may be another approach to disrupt proteostasis of MM cells. Recently, we found that the translation inhibitor omacetaxine is potent and efficacious in PI/IMiD refractory MM patient samples [100]. These strategies are still in early development for RRMM treatment, so further clinical trial results are needed to assess their potential to overcome PI resistance.

## 4. Immunotherapies

Immunotherapies have recently become a mainstay of MM treatment. Currently, three monoclonal antibodies (mAbs) are approved for MM treatment, and multiple next-generation immunotherapies are at various stages of clinical testing. mAbs are unique among anti-myeloma agents, acting to prime the immune system to specifically target malignant cells. In 2015, the FDA approved daratumumab monotherapy and elotuzumab-based combination treatment for RRMM. Daratumumab has since been approved in several combination regimens for newly diagnosed and RRMM, and elotuzumab in combination with dexamethasone and either lenalidomide or pomalidomide in RRMM. Daratumumab-based combinations have become highly important in treating RRMM, with the response rates being the highest in the relapsed setting to date [101]. Another anti-CD38 mAb, isatuximab, has also been approved for RRMM treatment in combination with pomalidomide and dexamethasone. Recently, the antibody-drug conjugate (ADC) targeting BCMA, belantamab mafodotin, received accelerated approval for RRMM. This appears to be the first of several new immunotherapies likely to be approved, as multiple chimeric antigen receptor (CAR) T-cell products, bispecific antibodies and antibody drug conjugates (ADCs) look promising in clinical trials.

### 4.1. Monoclonal Antibody Mechanism of Action

Daratumumab is a fully human mAb that binds to the surface ectoenzyme CD38, a PC marker highly expressed on MM cells [102]. It acts through multiple cytotoxic mechanisms, including antibody-dependent cell-mediated cytotoxicity (ADCC), complement-dependent cytotoxicity (CDC), antibody-dependent cellular phagocytosis (ADCP), and direct induction of cell death (Figure 3) [103,104]. Isatuximab targets a distinctly different CD38 epitope [105]. Elotuzumab targets the anti-signaling lymphocyte activation marker F7 (SLAMF7), which is expressed on more than 95% of MM cells [106]. SLAMF7 signaling also activates NK cell proliferation, so the elotuzumab effect may be accentuated through activation of NK cells.

### 4.2. Immunotherapy Resistance

Despite the efficacy of daratumumab, responders eventually develop resistance over time [102,107]. Intrinsic resistance appears due to low CD38 expression in some patients [25]. Acquired resistance is also expression-dependent, as daratumumab decreases CD38 levels even following the first infusion, and this decrease correlates with reduced CDC and ADCC at progression (Figure 3A) [25]. Interestingly, CD38 mutations in daratumumab-refractory patients have not been reported, although a limited number of daratumumab-treated patients have been sequenced [11]. Aside from CD38, upregulation of the complement inhibitors CD55 and CD59 also occurs in some patients at progression (Figure 3B), implying that may be a second event leading to daratumumab resistance [25]. CD38 loss appears to be transient, as patient’s MM cells recover their CD38 approximately six months after daratumumab discontinuation [25]. In agreement with this, ex vivo daratumumab sensitivity appears to be partially restored greater than one year after daratumumab discontinuation [108]. Clinically, recovering anti-CD38 mAb sensitivity could be of substantial benefit to patients. As elotuzumab has not shown single-agent activity, there are no resistance mechanisms currently known.

### 4.3. Overcoming Immunotherapy Resistance

Various next-generation immunotherapies are being investigated for MM, mostly targeting antigens other than CD38 or SLAMF7. These therapies feature formats with enhanced cytotoxicity and include chimeric antigen receptor (CAR)-T cells, antibody-drug conjugates (ADCs), and bispecific antibodies. The most frequently targeted alternative antigen for these immunotherapies has been B-cell maturation antigen (BCMA), which is expressed on MM cells in the vast majority of patients [109]. Prominent among the anti-BCMA therapies are CAR-T cells, engineered to express a chimeric T-cell receptor to recognize and facilitate the elimination of MM cells [57]. In a Phase II trial, the anti-BCMA CAR-T cell therapy idecabtagene vicleucel (ide-cel, formerly bb2121) showed a 73% (94/128) ORR in heavily-pretreated RRMM patients, including daratumumab-refractory patients [110]. Phase III results for ide-cel are not yet public, but have led to the recent FDA approval for this to be the first commercially available CAR-T cell product for MM. In addition, ciltacabtagene autoleucel (cilta-cel, formerly LCAR) has shown very promising results in a phase I/IIb single arm study, with an ORR of 95% (92/97) [58]. Bispecific antibodies are novel antibody therapeutics that have dual specificity for a target cancer cell antigen and CD3, by which they engage T cell mediated cytotoxicity [109]. The bispecific antibody CC-93269 showed favorable results in a Phase I trial for RRMM with an 83% (10/19) ORR [59]. In support of overcoming mAb resistance, 89% of patients were daratumumab-refractory. Other promising anti-BCMA CAR-T cells and bispecific antibodies for MM are also in development and are showing similarly high response rates (Table 2) [60,110,111,112]. ADCs are antibodies that deliver a cytotoxic payload directly to their target cancer cells. The anti-BCMA ADC belantamab mafodotin showed favorable results in a Phase I study that included 40% of patients that were daratumumab-refractory, and belantumab mafodotin was recently granted approval for RRMM [62]. In the near future, BCMA-targeted therapies are expected to drastically change the treatment landscape for RRMM.

Aside from BCMA-targeted immunotherapies, several other promising targeted immunotherapies are being pursued for MM treatment and may be especially useful for patients that relapse after BCMA-based therapy. CD46 is a complement inhibitor that is overexpressed in nearly all MM patients, and anti-CD46-ADC induced potent MM cell death in vitro and in vivo [113]. CD46-ADC is currently being evaluated in a Phase I trial for RRMM (Table 2). The bispecific antibody talquetamab is currently in a Phase I trial for RRMM. This bispecific targets the G protein-coupled receptor C5D (GPRC5D) and CD3 [114]. GPRC5D is expressed on myeloma cells but not regular PCs, making it an attractive therapeutic target. GPRC5D is also a potential target for CAR-T cell therapy [115]. The bispecific antibody cevostamab (BFCR4350A), targeting FCRH5 and CD3, is another bispecific antibody under clinical evaluation for MM [61]. The large repertoire of immunotherapies currently being evaluated in RRMM will offer an array of highly specific therapies to target MM cells, and in combination with other anti-MM agents may elicit deeper and more durable therapeutic responses.

## 5. Other Drug Classes

In addition to the three aforementioned drug classes, glucocorticoids and chemotherapeutics also remain vital components of myeloma treatment in both newly-diagnosed and RRMM. Dexamethasone is the most commonly used glucocorticoid in MM, and melphalan is a frequently utilized alkylating chemotherapeutic agent that also possesses anti-MM immunostimulatory effects [2,26].

### 5.1. Dexamethasone Resistance

All glucocorticoids target the glucocorticoid receptor (GR) and binding leads to nuclear translocation and changes in gene expression. GRs have many downstream targets, ultimately repressing anti-apoptotic and metabolic pathways that drive MM cell death [26]. In MM and other hematologic malignancies where steroids are commonly used, the most common resistance mechanism is decreased GR expression, and low levels correlate with poorer outcomes. Mutations and alternatively-spliced isoforms of GR have also been reported, but these appear to be rare [11]. Studies have also suggested that certain subgroups of MM respond better to dexamethasone [116]. Dexamethasone sensitivity in RRMM needs to be further investigated, as it is used in almost all lines of therapy.

### 5.2. Melphalan Resistance

As with chemotherapeutic agents in general, increased expression of drug efflux pumps, such as P-gp and other multidrug resistance (MDR) proteins, have been implicated in MM melphalan resistance [27]. Since alkylating agents primarily act through inducing irreparable DNA damage, it follows that increased expression of factors related to DNA repair or replication, such as RECQ1 (encoding the RecQ1 helicase), are associated with melphalan resistance in MM [28,117]. Additionally, high expression of anti-apoptotic genes, including BCL-xL, have also been associated with melphalan resistance [27]. While agents that target P-gp have been largely unsuccessful, agents that further exacerbate DNA damage, or prevent escape from apoptosis, may accentuate alkylator efficacy in RRMM.

### 5.3. Overcoming Melphalan Resistance

Melflufen (Melphalan-flufenamide) is a first-in-class peptide-drug conjugate that was recently FDA-approved for triple-class refractory MM. It was designed to selectively deliver melphalan to MM cells [118]. Due to its lipophilic nature, melflufen is rapidly taken up by MM cells, and once inside the cell it is rapidly cleaved by peptidases to release melphalan [66]. This accelerated uptake and release likely explain the increased efficacy of melflufen compared to melphalan. When evaluated in a Phase II trial, melflufen had an ORR of 26% (31/119) in triple-refractory patients [66]. Melflufen is also being assessed in combination with dexamethasone and either bortezomib or daratumumab in RRMM (Table 2).

## 6. Other Strategies for Circumventing Drug Resistance

In addition to improvements within the above classes, other drugs are under development for RRMM treatment. These agents have distinct mechanisms of action and may better avoid cross-resistance with routinely used drugs. Selinexor is the best example of this, having received accelerated approval in the context of triple-class refractory MM patients. Selinexor is a SINE compound that inhibits Exportin 1 (XPO1), a nuclear export protein that controls the localization of many proteins important for MM survival [119]. Selinexor showed an ORR of 26% (32/122) in PI/IMiD triple-refractory patients in a Phase I/II study in combination with dexamethasone (Table 2) [120]. In addition, selinexor with bortezomib and dexamethasone showed an ORR of 76% (149/195), with a progression free survival of 13.9 months in a Phase III trial of myeloma patients who had been treated with one to three prior lines of therapy [121].

A more recent approach in anti-cancer therapeutics is to inhibit the anti-apoptotic proteins cancer cells exploit. MM shows a heterogenous dependency on the anti-apoptotic B-cell lymphoma 2 (BCL-2) family members BCL-2, BCL-xL, and MCL-1 [122]. Venetoclax is a BH3 mimetic that selectively inhibits BCL-2 and is approved for use in chronic lymphoid leukemia (CLL) and acute myeloid leukemia (AML). Early clinical trials in MM suggested that venetoclax synergizes with PIs [123]. In a phase III clinical trial in RRMM, venetoclax addition to bortezomib and dexamethasone led increased progression free survival, but unfortunately also increased mortality, leading to a FDA-imposed halt of the study [67]. However, preclinical and clinical results strongly support that venetoclax activity in MM is associated with t (11;14), leading to the possibility those specific patients would have a better risk-benefit ratio [67,122,124]. Thus, clinical study of venetoclax in RRMM is currently focused on t (11;14)-positive disease (Table 2).

## 7. Conclusions and Future Directions

Although MM has seen a series of therapeutic revolutions over the past two decades that have drastically improved clinical outcomes, patients still inevitably develop multi-drug resistance and ultimately succumb to their disease. Thus, the triple-class refractory setting is currently the greatest area of clinical need in MM. Understanding drug resistance mechanisms is critical for identifying novel and effective drug targets and designing rational therapeutic combinations for triple-class refractory patients. New drug classes that target these mechanisms or drugs within the same class that do not show cross-resistance are fitting strategies. However, drug resistance in MM, like the disease itself, is complex and heterogenous, so it is unlikely one mechanism alone explains every case of resistance to that drug class. Pinpointing drug resistance mechanisms in patients is further convoluted by the fact that patients nearly always receive combination therapy, so it is difficult to say exactly which drug(s) a patient is actually resistant to in the combination. This is exacerbated furthermore by the widespread occurrence of multi-drug resistance, as some of these agents work synergistically or have overlapping pathways involved in their mechanisms of action. It is possible that the best way to evaluate drug resistance will be on a patient-to-patient basis.

One feasible approach to complement drug development may be personalized medicine (also known as precision medicine). This would involve identifying reliable predictive biomarkers of response and profiling individual patients for what agents are likely to elicit a successful therapeutic response, which would inform effective treatment regimens. Approaches to accomplishing this could include predicting patient drug response based on their genetic profile or to profile patient’s sensitivity to anti-myeloma agents ex vivo over their disease course [54,125,126]. Several of these approaches are currently being assessed in clinical trials for MM. Previous studies have suggested that resistance to one drug class reprograms resistant clones in a manner that yields them increased sensitivity to other drug classes or to drugs within the same class [13,127]. It is also possible that patients may become re-sensitized to an agent as they are exposed to other lines of treatment, so this could also help guide if agents can be reintroduced in patients with recaptured efficacy at a later point in their disease.

The observation of more progenitor-like MM cells in PI resistant cell lines and patient samples also raises the question of whether our standard approaches to characterizing MM are robust/conserved enough to capture the full disease, especially in the relapsed/refractory setting. MM is generally characterized by double-positivity for the PC markers CD38 and CD138; however, studies have suggested that MM differentially express CD138 or downregulate CD138 expression [128,129]. Additional cellular markers of MM that allow us to account more accurately for the disease heterogeneity, such as CD46 or BCMA, should be considered when studying MM, of which next-generation flow and mass cytometry that allow for a greater number of cellular markers will be instrumental.

Continuing to expand our understanding of the complexities of drug resistance mechanisms will identify novel targets and strategies that enable us to overcome resistance and offer reliable predictive biomarkers. These discoveries could guide clinical decisions for multiple myeloma treatment and may potentially even lead to a cure in some patients.

## Figures and Tables

**Figure 1 cancers-13-01686-f001:**
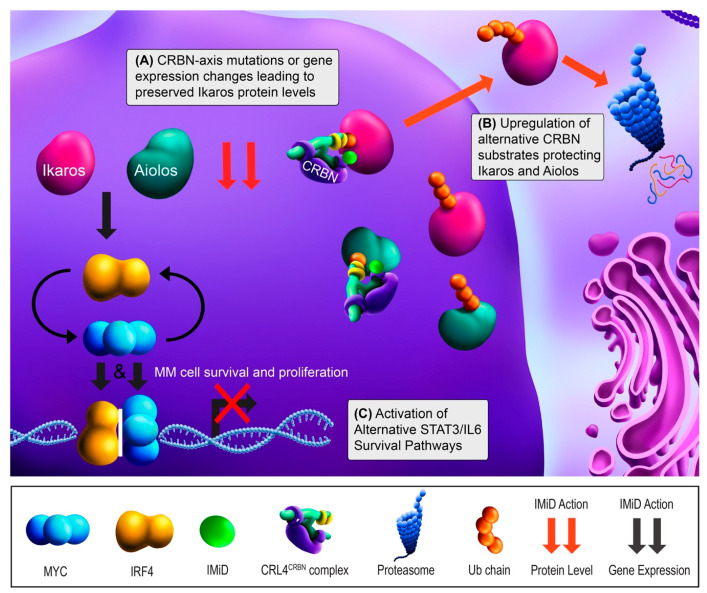
Immunomodulatory drug (IMiD) mechanism of action and resistance mechanisms. Ikaros (IKZF1) and Aiolos (IKZF3) increase the gene expression of IRF4 (yellow) and MYC (cyan), leading to multiple myeloma (MM) cell survival and proliferation. IRF4 and MYC bind to their respective promoters independently, but also upregulate each other’s gene expression. With treatment, IMiD binding leads to the Cereblon (CRBN)-dependent proteasomal degradation of ubiquitin (Ub)-tagged Ikaros and Aiolos (red arrows). IKZF1/3 degradation leads to the downregulation of IRF4 and MYC gene expression (black arrows) and results in growth arrest and apoptosis (red “X”). (**A**) IMiD resistance can occur when components in the CRBN-Ikaros axis are mutated or undergo altered expression, most frequently downregulation of CRBN. (**B**) IMiD resistance can also occur through increased expression of alternative CRBN substrates that effectively prevent IKZF1/3 degradation by binding CRBN or by mediating the CRBN degradation. (**C**) Finally, IMiD resistance can occur via upregulation of alternative survival pathways to bypass Ikaros axis dependence, including IL-6/STAT3 activation.

**Figure 2 cancers-13-01686-f002:**
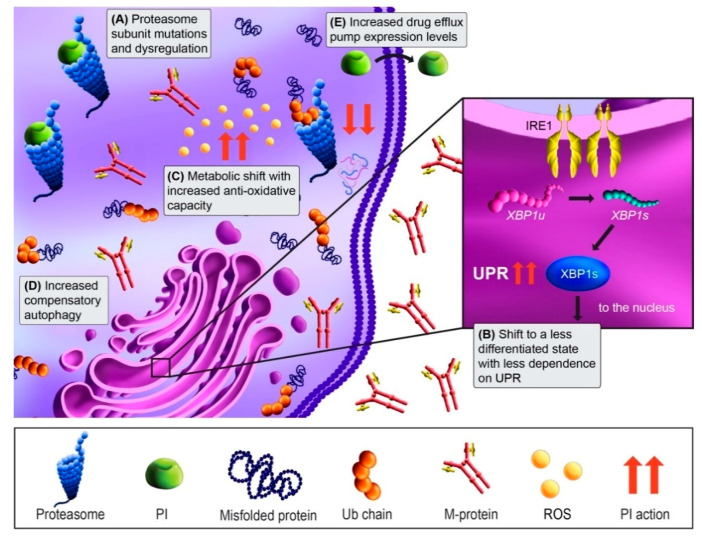
Proteasome inhibitor (PI) mechanism of action and resistance mechanisms. MM cells are dependent on the proteasome to degrade and recycle the high level of misfolded proteins that result from the production of M-protein. Treatment with PIs (green) blocks the degradation of ubiquitinated (Ub) proteins by binding either reversibly (bortezomib) or irreversibly (carfilzomib) to the proteasome. This blocks protein degradation, leading to the accumulation of misfolded proteins and activation of the unfolded protein response (UPR) and apoptosis (red arrows). (**A**) PI resistance can occur through mutations in proteasome catalytic subunits, or via dysregulation of subunit expression. (**B**) PI resistance may also occur in MM cells through a shift to a less differentiated phenotype that produces less M-protein and thus exhibit less dependence on the IRE1-Xbp1s UPR arm. (**C**) In addition, a shift in metabolic state can occur with PI resistance that more effectively neutralizes the cytotoxic reactive oxygen species (ROS) that are generated by UPR. (**D**) Autophagy upregulation is another compensatory adaptation that can maintain proteostasis when the proteasome is inhibited. (**E**) PI-resistant MM cells may adapt to PI treatment via upregulation of drug efflux pumps that extrude the drugs from the cells.

**Figure 3 cancers-13-01686-f003:**
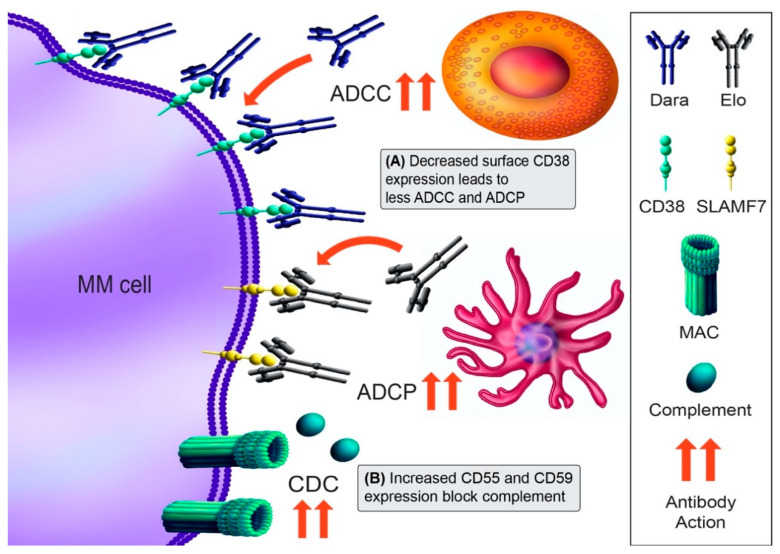
Immunotherapy mechanisms of action and resistance mechanism. The monoclonal antibody (mAb) daratumumab (Dara) binds to CD38 and induces cytotoxicity through antibody-dependent cellular cytotoxicity (ADCC; NK cell [orange], macrophage [pink]), complement-dependent cytotoxicity (CDC; membrane attack complex [MAC]), antibody-dependent cellular phagocytosis (ADCP; macrophage), and direct blocking of CD38 function. Elotuzumab (Elo) recognizes SLAMF7 and results in direct activation of NK cells and ADCC. (**A**) Downregulation of CD38 mediates Dara resistance, in part by trogocytosis of CD38 from MM cells. (**B**) A second mechanism of Dara resistance may be the upregulation of complement-inhibitory proteins CD55 and CD59. Antibody action is denoted by red arrows. The mechanisms shown are for Dara resistance mechanisms, as no Elo resistance mechanisms have been described.

**Table 1 cancers-13-01686-t001:** Resistance mechanisms to targeted therapies in multiple myeloma.

Drug Class	Drug Name	Drug MOA	Resistance Mechanisms	Refs
Immunomodulatory Drugs (IMiDs)	Thalidomide Lenalidomide	CRBN-dependent degradation of IKZF1/3, immune modulation, anti-angiogenic/inflammatory	CRBN-Ikaros axis mutations/transcriptional regulation, IKZF1/3 protection, upregulation of IL-6/STAT3 pathway	[8,9,10,11,12,13,14,15,16,17]
	Pomalidomide			
Proteasome Inhibitors (PIs)	Bortezomib Ixazomib	Inhibit 26S proteasome though reversibly binding the PSMB5 subunit	Proteasome subunit mutations or upregulation, de-differentiation, alternate proteostasis pathways, increased drug efflux	[18,19,20,21,22,23,24]
	Carfilzomib	Same as others, but binds irreversibly, also binds to PSMB2		
Monoclonal Antibodies	Daratumumab Isatuximab Elotuzumab	CDC, ADCC, ADCP, for CD38 antibodies, direct induction of cellular apoptosis	Decreased target expression, increased expression of complement inhibitory proteins	[25]
Glucocorticoids	Dexamethasone	Repress anti-apoptotic and metabolic pathways	Decreased GR expression, GR mutations	[26]
Chemotherapies	Melphalan	Causes DNA damage through alkylation	Increased drug efflux, increased expression of DNA repair factors	[27,28]

MOA = mechanism of action, CDC = complement-dependent cytotoxicity, ADCC = antibody-dependent cellular cytotoxicity, ADCP = antibody-dependent cellular phagocytosis, GR = glucocorticoid receptors.

**Table 2 cancers-13-01686-t002:** Key therapies currently in clinical trials for overcoming drug resistance in multiple myeloma.

Current MM Drug Class	New Drug Class	Drug Name	Drug Targets	Clinical Trial Status (Clinical Trial #)	Refs
Immunomodulatory Drugs (IMiDs)	CELMoDs	Iberdomide (CC-220) Avadomide (CC-122) CC-92480	Cereblon	Phase I/II (NCT02773030) Phase I (NCT01421524)	[52,53]
				Phase I (NCT03374085)	
Proteasome Inhibitors (PIs)	PIs	Marizomib (NPI-0052)	β5, β2, and β1 subunits of 20S proteasome	Completed Phase II (NCT00461045)	[55]
	Autophagy inhibitor	Hydroxychloroquine	Autolysosome formation	Completed Phase I (NCT00568880)	[56]
Monoclonal Antibodies (mAbs)	CAR-T cells	Ide-Cel (bb2121)	BCMA	FDA approved for RRMM Phase III (NCT03361748)	[57,58]
		Cilta-Cel (JNJ-4528)		Phase III (NCT04181827)	
		BCMA-CART		Phase I (NCT02215967)	
	Bispecific antibodies	REGN5458	BCMAxCD3	Phase I/II (NCT03761108)	[59,60,61]
		CC-93269		Phase I (NCT03486067)	
		Teclistamab		Phase II (NCT04557098)	
		PF-3135		Phase I (NCT03269136)	
		AMG 701		Phase I (NCT03287908)	
		TNB-383B		Phase I (NCT03933735)	
		Talquetamab	GPRC5DxCD3	Phase I (NCT03399799)	
		Cevostamab	FCRH5xCD3	Phase I (NCT03275103)	
	Antibody-drug conjugates	Belantamab mafodotin (GSK2857916)	BCMA	FDA approved for RRMM	[62,63]
		FOR46	CD46	Phase I (NCT03650491)	
Other	SINE	Selinexor	XPO1	FDA approved for RRMM	[64]
	HDAC inhibitors	Ricolinostat	HDAC6	Phase I/II (NCT01997840)	[65]
	Peptide-drug conjugates	Melflufen	Plasma cells (lipophilic cells)	FDA approved for RRMM Phase II (NCT02963493)	[66]
	BH3 mimetics	Venetoclax	BCL-2	Phase III (NCT2755597); t (11;14) RRMM Phase I/II (NCT01794520)	[67]

CELMoDs = Cereblon modulators, CAR = chimeric antigen receptor, SINE = selective inhibitor of nuclear export, HDAC = histone deacetylase. Clinical Trial # = ClinicalTrials.gov Identifier.

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
