# Peer review of "Emerging Therapeutic Strategies to Overcome Drug Resistance in Multiple Myeloma"

_cancers, 2021, doi:10.3390/cancers13071686_

Round 1

Reviewer 1 Report

This review by Davis and Sherbenou lays out the major classes of anti-myeloma therapeutics, their mechanism of action and ongoing efforts to develop novel therapies to overcome the inevitable resistance. It is comprehensive, well written and nicely organized, and I believe it will be well received by the myeloma and heme malignancy communities. I have relatively minor points listed below to fix typos, clarify parts of the text or small modifications to include important points.

Minor points:

  • Line 9: “largely incurable”. I think we can say that it’s incurable.
  • Line 10: “acquired” should be “acquire”
  • Line 42: typo “extraordinary”
  • Line 45: There is a missing word after “accumulation of genetic”
  • Line 55: Now with the approval of isatuximab, I would suggest using anti-CD38 instead of daratumumab.
  • Table 1: Should include thalidomide among the IMiDs. It’s an approved agent that is still used around the world, and in certain circumstances in the US. Would also include ixazomib (chemically very similar to bortezomib) among the PI’s, and isatuximab and elotuzumab among the MoAbs.
  • Line 92: I don’t believe there’s any evidence that IMiDs induce a conformational change in CRBN. Instead, they bind to CRBN and act as a molecular glue between CRBN and substrate proteins.
  • Section 2.1: I would reframe this as the bulk of the data suggests that IRF4/Myc are the key hub in myeloma and other mature B-cell malignancies. Any way that you target them leads to PC cell death. Ikaros/Aiolos degradation is just one way of doing that.
  • Line 104: These numbers are for RRMM. The ORR is much higher in newly diagnosed patients and when combined with dex.
  • Line 122: I think it depends what you consider rare. Once you begin to add up mutations in all the various components of the E3 ligase machinery (CUL4A/B, CRBN, DDB1, etc) it may be that genetic alterations in this pathway are not that uncommon. The recent work by Gooding et al (https://doi.org/10.1182/blood.2020007081) should also be cited here.
  • Section 2.3: The fact that from what we know almost all next generation IMiDs (from pomalidomide to CC92480) work by being more potent degraders of IKZF1/3 also reinforces the idea that much of clinical resistance to Lenalidomide is due to downregulation or loss of CRBN or related components.
  • Line 189: It’s unclear to me why MEK/ERK inhibition would be particularly effective in IMiD resistant patients. MAPK activating mutations are common in myeloma and this pathway may be targetable but that would be independent of IMiDs. I would suggest removing this.
  • Figure 1: The figure implies that MYC and IRF4 dimerize at promoters, but I’m not aware of any such action.
  • Section 3.3: Given that panobinostat was developed partially because of preclinical data suggesting that it could help overcome bortezomib resistance it may be worth discussing that data in this section.
  • Line 370: The phase 2 data has recently been published in NEJM and can be cited here. It may be worth mentioning that FDA approval of bb2121 is expected imminently (likely before the publication of this review). I would also mention the excellent data for Cilta-cel, since it will also likely be approved in the coming year.
  • Section 4.3: It might be worth having a brief discussion of other immune-targeted agents including checkpoint inhibitors (PD-1, PD-L1, LAG3, CD47, etc). The results have been mixed to bad so far, but there may still be hope for combinations in the future to help overcome resistance. Could also mention the use of gamma-secretase inhibitors to upregulate BCMA.
  • Section 5.3: Would mention the recent FDA approval of melflufen.
  • Line 489: Could also mention other approaches to ex vivo drug sensitivity measurements such as Cetin et al (https://doi.org/10.1038/s41467-017-01593-2) using mass accumulation rate, or BH3 profiling: Chongaile et al (https://doi.org/10.1126/science.1206727).
  •  

Author Response

Dear Reviewer 1, 

Thank you for your helpful and thorough evaluation of our manuscript. We appreciate your time. We have addressed each comment below, with our responses to each point denoted by italic text. The corresponding changes made in the revised manuscript are highlighted in red text. 

Minor points:

  • Line 9: “largely incurable”. I think we can say that it’s incurable. Response: We agree, although allogeneic transplant can be curative for myeloma, the risk currently outweighs the benefits with the wealth of good treatment options available. This change has been made. 
  • Line 10: “acquired” should be “acquire”. Response: Corrected, thank you.
  • Line 42: typo “extraordinary”. Response: Corrected, thank you.
  • Line 45: There is a missing word after “accumulation of genetic”. Response: the word "mutations" has been added.
  • Line 55: Now with the approval of isatuximab, I would suggest using anti-CD38 instead of daratumumab. Response: We agree, this has been changed.
  • Table 1: Should include thalidomide among the IMiDs. It’s an approved agent that is still used around the world, and in certain circumstances in the US. Would also include ixazomib (chemically very similar to bortezomib) among the PI’s, and isatuximab and elotuzumab among the MoAbs. Response: Thank you for the suggestion, these agents have been added to the table.
  • Line 92: I don’t believe there’s any evidence that IMiDs induce a conformational change in CRBN. Instead, they bind to CRBN and act as a molecular glue between CRBN and substrate proteins. Response: We agree, this has been changed.
  • Section 2.1: I would reframe this as the bulk of the data suggests that IRF4/Myc are the key hub in myeloma and other mature B-cell malignancies. Any way that you target them leads to PC cell death. Ikaros/Aiolos degradation is just one way of doing that. Response: We agree, this paragraph has been adjusted to focus on IRF4 and MYC as the ultimate targets of IMiD effects in MM.
  • Line 104: These numbers are for RRMM. The ORR is much higher in newly diagnosed patients and when combined with dex. Response: This is a good point, we have rephrased the sentence to reflect your comment. 
  • Line 122: I think it depends what you consider rare. Once you begin to add up mutations in all the various components of the E3 ligase machinery (CUL4A/B, CRBN, DDB1, etc) it may be that genetic alterations in this pathway are not that uncommon. The recent work by Gooding et al (https://doi.org/10.1182/blood.2020007081) should also be cited here. Response: We agree, we have modified this sentence has been changed, removing the word "rare" and we have added the citation for Gooding et al. Thank you for bringing up their work. 
  • Section 2.3: The fact that from what we know almost all next generation IMiDs (from pomalidomide to CC92480) work by being more potent degraders of IKZF1/3 also reinforces the idea that much of clinical resistance to Lenalidomide is due to downregulation or loss of CRBN or related components. Response: We agree with your comment and have added a sentence to this section to reflect this point. 
  • Line 189: It’s unclear to me why MEK/ERK inhibition would be particularly effective in IMiD resistant patients. MAPK activating mutations are common in myeloma and this pathway may be targetable but that would be independent of IMiDs. I would suggest removing this. Response: Thank you for the comment. We agree and have removed this. 
  • Figure 1: The figure implies that MYC and IRF4 dimerize at promoters, but I’m not aware of any such action. Response: We didn't mean to imply dimerization. We have updated Figure 1 and its legend to reflect the independent actions on gene expression of IRF4 and MYC. 
  • Section 3.3: Given that panobinostat was developed partially because of preclinical data suggesting that it could help overcome bortezomib resistance it may be worth discussing that data in this section. Response: Thank you for the comment, this addition has been made. 
  • Line 370: The phase 2 data has recently been published in NEJM and can be cited here. It may be worth mentioning that FDA approval of bb2121 is expected imminently (likely before the publication of this review). I would also mention the excellent data for Cilta-cel, since it will also likely be approved in the coming year. Response: Thank you for making these points. We have replaced the phase 1 data for bb2121 with phase II, and have noted the recent FDA-approval of this cell therapy. We have also noted the promising trial results with cilta-cel with the appropriate citation. 
  • Section 4.3: It might be worth having a brief discussion of other immune-targeted agents including checkpoint inhibitors (PD-1, PD-L1, LAG3, CD47, etc). The results have been mixed to bad so far, but there may still be hope for combinations in the future to help overcome resistance. Could also mention the use of gamma-secretase inhibitors to upregulate BCMA.
  • Section 5.3: Would mention the recent FDA approval of melflufen. Response: This has been noted, thank you.
  • Line 489: Could also mention other approaches to ex vivo drug sensitivity measurements such as Cetin et al (https://doi.org/10.1038/s41467-017-01593-2) using mass accumulation rate, or BH3 profiling: Chongaile et al (https://doi.org/10.1126/science.1206727). Response: We appreciate your comment. The reason we had mentioned the work from our lab (Walker et al. Blood Advances, 2020) in the context of this review, is that is has been the only platform to this point that can measure drug sensitivity to all three drug classes focused on in this manuscript (IMiDs, PIs and mAbs). Respectfully, we have not cited other ex vivo assays for drug sensitivity in the revision for this reason. 

Reviewer 2 Report

I think this is a very well-written and comprehensive review on anti-MM drugs and mechanisms of resistance. I will only add two parts in the lenalidomide session:

  1. It has been reported that lenalidomide also modulates phosphorylation of TP53RK, by direct binding. (Hideshima, Blood, 2017). I will include a paragraph in the "IMID Mechanisms of action" session. 
  2. The authors state that CRBN mutations are rare. However, using WGS and RNA-seq  a recent Blood paper showed that the overall incidence of CRBN alterations (copy loss, mutation, alternative splicing) is >20% in len/pom resistance patients (Gooding et al, 2020; "Multiple cereblon genetic changes are associated with acquired resistance"). This needs to be included and discussed. 

Author Response

Dear Reviewer 2, 

Thank you for your helpful evaluation of our manuscript. We appreciate your time. We have addressed each comment below, with our responses to each point denoted by italic text. The corresponding changes made in the revised manuscript are highlighted in red text. 

I think this is a very well-written and comprehensive review on anti-MM drugs and mechanisms of resistance. I will only add two parts in the lenalidomide session:

  1. It has been reported that lenalidomide also modulates phosphorylation of TP53RK, by direct binding. (Hideshima, Blood, 2017). I would include a paragraph in the "IMID Mechanisms of action" session. Response: As this currently stands alone as the only paper making a connection of TP53RK to IMiD mechanism of action, we feel the connection is less supported at this time.
  2. The authors state that CRBN mutations are rare. However, using WGS and RNA-seq  a recent Blood paper showed that the overall incidence of CRBN alterations (copy loss, mutation, alternative splicing) is >20% in len/pom resistance patients (Gooding et al, 2020; "Multiple cereblon genetic changes are associated with acquired resistance"). This needs to be included and discussed. Response: We have changed lines 122-123, 33, and 175, and have added the Gooding et al. reference.

Reviewer 3 Report

very nice paper

excellent timing for a paper of this nature

consider discussing the role of HDACi overcoming resistance as this has been shown both in vivo and in clinical studies

Author Response

Dear Reviewer 3, 

Thank you for your helpful evaluation of our manuscript. We appreciate your time. We have addressed each comment below, with our responses to each point denoted by italic text. The corresponding changes made in the revised manuscript are highlighted in red text. 

Consider discussing the role of HDACi overcoming resistance as this has been shown both in vivo and in clinical studies. Response: We have added HDACi to the section on overcoming PI resistance (lines 320-326).